# LG-BENCH: A GRAPH-STRUCTURED EVALUATION BENCHMARK FOR LIFE SCIENCES

## ABSTRACT

Traditional evaluation benchmarks reduce inherently interconnected scientific knowledge in life sciences into flat lists of questions, disregarding the underlying topological structure of the knowledge. We introduce, the first graph-structured benchmark for life sciences, featuring over 10,000 high-quality multiple-choice questions across medicine, biology, and chemistry. Our approach constructs a weighted evaluation graph using bidirectional matching and semantic similarity algorithms, where nodes represent questions and edge weights capture their semantic relationships. Leveraging this graph topology, we design two novel evaluation metrics. The Global Coherence Score (GCS) measures a model's consistency within semantically related neighborhoods, while Knowledge Balance Score (KBS) analyzes how model errors are distributed across the graph to reveal conceptual blind spots. LG-Bench facilitates fine-grained comparison of LLMs by surfacing differences in conceptual coherence and patterns of knowledge organization across models. Our framework shifts the evaluation paradigm from flat accuracy metrics to structure-aware analysis, offering a new lens for diagnosing and improving LLM performance in the life sciences domain.

## 1 INTRODUCTION

Recent breakthroughs in large language models (LLMs) epitomized by systems such as GPT-4 (Team, 2024), and Llama (Meta AI, 2024) have rapidly accelerated progress in natural language processing and sparked an intense wave of research activity across multiple scientific disciplines. As model scale and performance continue to increase, the community is increasingly dependent on systematic evaluation to guide development, compare approaches, and guarantee safe deployment.

However, current evaluations such as practices in artificial intelligence suffer from a fundamental misrepresentation: they treat knowledge as isolated, independent facts when it naturally forms interconnected webs of understanding. This "flat-world illusion" is particularly problematic in complex domains such as life sciences (Bodenreider, 2004; Jin et al., 2021), where understanding protein function requires grasping its role in cellular pathways, regulatory mechanisms, and therapeutic implications as an integrated whole. Traditional benchmarks present evaluation questions as flat lists, obscuring the rich semantic relationships that define genuine domain expertise. These limita-

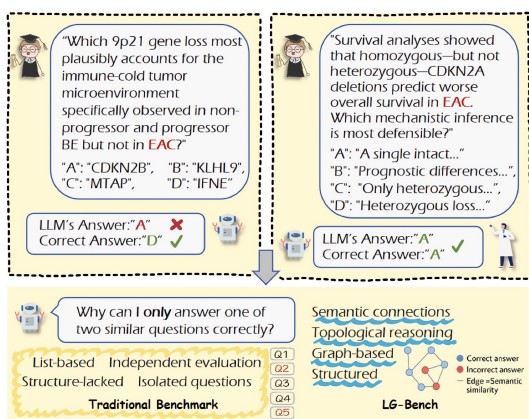

Figure 1: A case study highlights the limitations of traditional benchmarks and the advantages of LG-Bench. Despite high semantic similarity between two related questions, large language models often answer only one of them correctly. Traditional benchmarks use a list-based structure, making it difficult to uncover semantic connections between questions. LG-Bench structures questions as a graph, enabling models to analyze semantic links and allowing for more effective evaluation of their capabilities.

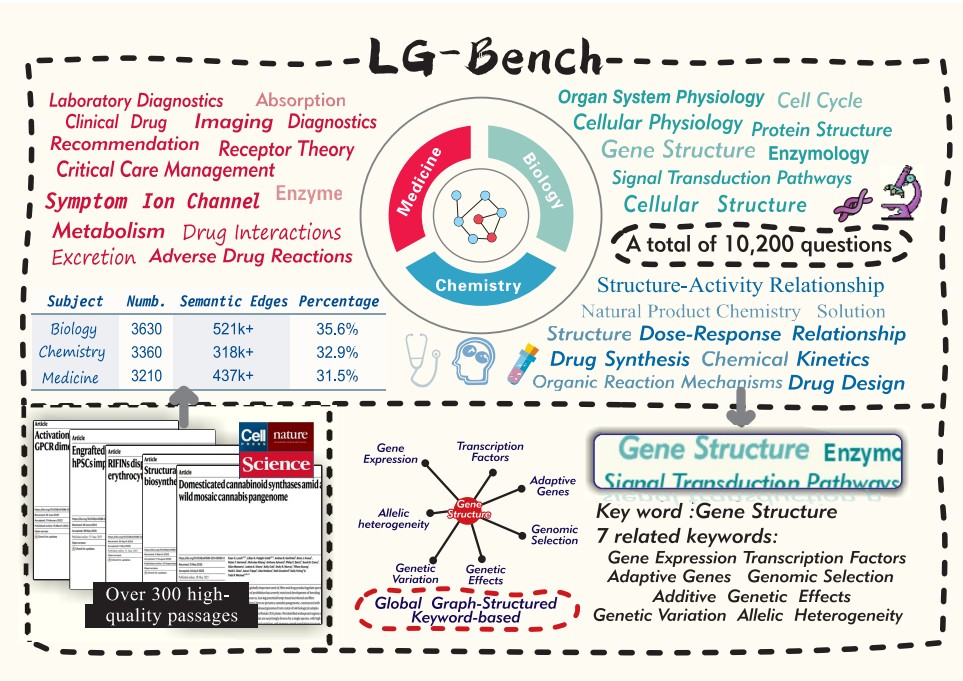

Figure 2: Overview diagram of LG-Bench. Our LG-Bench extracts over 10,000 questions from top-tier journals in the life sciences, covering three subfields: Medicine, Biology, and Chemistry. It innovatively introduces graph structures into the benchmark to better capture the nuanced capabilities of large language models.

tions make it hard for current metrics to distinguish between genuine understanding and mere pattern matching.

To solve this problem, we introduce the first graph-structured benchmark for comprehensive life sciences evaluation. Unlike traditional flat benchmarks that treat questions as isolated units, our approach reveals and exploits the inherent evaluation graph structure in evaluation datasets, as shown in Fig. 1. We construct a large-scale benchmark containing over 10k expert-reviewed questions drawn from recent peer-reviewed scientific literature, spanning diverse life sciences domains including Medicine, Biology, and Chemistry. Representative sources include Nature, Science, and Cell, ensuring coverage of both foundational knowledge and cutting-edge research. Through sophisticated graph construction methods using bidirectional matching and semantic similarity, we transform this question corpus into a weighted evaluation graph that captures the interconnected nature of scientific knowledge. This graph structure enables novel evaluation approaches that distinguish true understanding from superficial pattern matching by measuring knowledge coherence and analyzing the topological distribution of model errors across semantically related questions.

Our contributions are threefold:

- We introduce the first graph-structured benchmark for life sciences, containing over 10k expert-curated questions from recent literature from leading scientific journals spanning Medicine, Biology, and Chemistry, as shown in Fig. 2.

- We propose novel graph-based evaluation methods including the Global Coherence Score (GCS) for measuring knowledge coherence and the Knowledge Balance Score (KBS) for quantifying variance in local coherence patterns across the knowledge graph topology.

- We conducted a systematic evaluation of LG-Bench across large language models of varying scales, and leveraged GCS and KBS to deeply analyze performance flaws, providing practical support for capability diagnosis and optimization of large models in the life science.

## 2 RELATED WORK

Traditional evaluation of large language models relies on accuracy-based metrics, treating each question as an independent unit (Li et al., 2024; Richard, 2015). This approach assumes knowledge can be divided into isolated facts, missing the interconnected nature of true expertise. Although recent work explores more nuanced frameworks (Mondorf & Plank, 2024; Xu et al., 2025; Zhang et al., 2025), current metrics still struggle to differentiate genuine understanding from sophisticated pattern matching, a limitation particularly evident in complex scientific domains.

This challenge is amplified in the life sciences. Prominent benchmarks like UMLS (Bodenreider, 2004), MedQA (Jin et al., 2021), PubMedQA (Jin et al., 2019), MLEC-QA (Li et al., 2021), and BioASQ (Nentidis et al., 2023) suffer from critical limitations. They often rely on static knowledge, failing to capture the rapid evolution of research in areas like personalized medicine and advanced therapeutics (Cai et al., 2024; Chen et al., 2025; Zhou et al., 2025). Furthermore, their narrow focus on medicine over foundational sciences like biology and chemistry restricts the ability to assess comprehensive, interdisciplinary knowledge, which is essential for reliable evaluation.

A promising direction to address these structural flaws lies in graph-based knowledge representation. While knowledge graphs are extensively studied in NLP (Hogan et al., 2021; Ji et al., 2021), and techniques like graph embeddings (Bordes et al., 2013; Wang et al., 2017) and graph neural networks (Kipf, 2016; Wu et al., 2020) have proven effective at capturing complex relationships, these insights have not yet been applied to benchmark construction. In the life sciences, where knowledge inherently forms interconnected networks from molecular interactions to physiological systems, the absence of a graph-structured evaluation framework represents a significant gap. Our work bridges this gap by introducing the first graph-structured benchmark that explicitly models and leverages these knowledge relationships for a more meaningful and robust evaluation.

## 3 GRAPH-STRUCTURED BENCHMARK CONSTRUCTION

Our benchmark construction follows a two-stage pipeline shown as Fig. 3. First, we generate high-quality evaluation questions from recent scientific literature through a sophisticated multi-stage process with expert validation. Second, we construct a weighted evaluation graph, transforming the flat question corpus into a structured representation that reflects the interconnected nature of scientific knowledge.

### 3.1 QUESTION NODE GENERATION

We generate question sets through a sophisticated multi-stage pipeline that transforms cutting-edge scientific literature into high-quality evaluation questions.

**Document Analysis.** Our pipeline begins with a multi-modal large language model (LLM) analyzer that processes recent peer-reviewed papers from leading journals in the life sciences. This analyzer extracts structured knowledge representations from each document $d \in D_{\text{corpus}}$, identifying core concepts along with their hierarchical and semantic relationships. It systematically extracts knowledge pairs, consisting of concepts and their associated properties or relations, which serve as fundamental units for downstream question generation. The analyzer also assigns each identified concept a corresponding Bloom's Taxonomy level (Forehand, 2010), specifying its cognitive depth. The system enforces explicit targets for the distribution of questions across these levels, ensuring a balanced coverage of cognitive complexity in the final benchmark.

**Guided Question Generation.** The generation stage employs a specialized LLM that receives guidance from the analyzer's output. Each generation prompt includes:

- Domain-specific knowledge context and key concepts

- Target Bloom level with specific cognitive verbs

- Scientific accuracy constraints

- Requirements for testing conceptual relationships

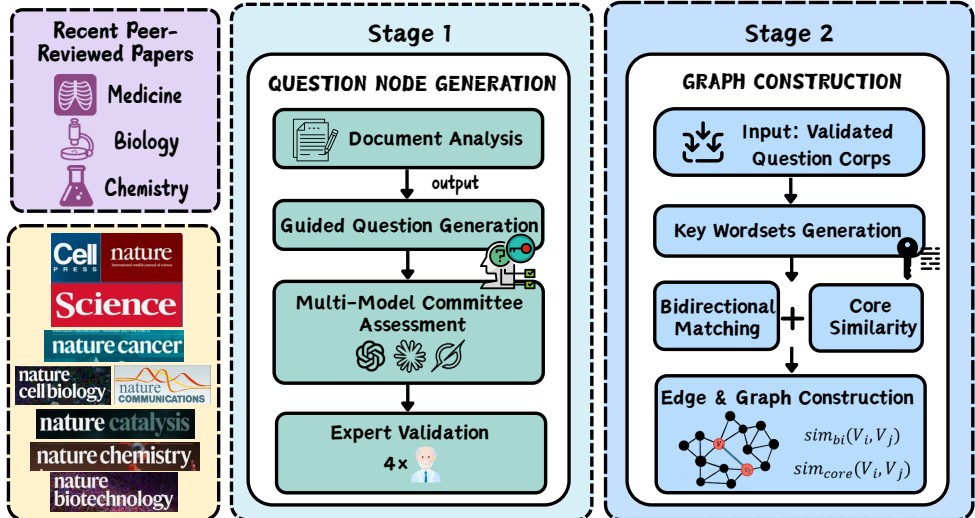

Figure 3: Two-stage Pipeline for Construction of LG-Bench.

This guidance ensures generated questions genuinely test integrated understanding rather than isolated fact recall.

**Multi-Model Committee Assessment.** The quality assurance stage involves a committee of three state-of-the-art large language models (GPT-4o, Claude 3.7 Sonnet, and Grok-3) that independently evaluate each generated question. The committee members assess questions across multiple dimensions, including scientific accuracy, cognitive complexity, and conceptual integration. Through a voting mechanism, questions that fail to meet quality thresholds are flagged for mandatory human expert review. Additionally, the committee annotates the relevant knowledge sources and the rationale behind each judgment to facilitate more efficient and accurate expert verification.

**Expert Validation.** Our human validation team comprises 4 PhD-holding life sciences experts. These experts conduct rigorous review by simultaneously examining both the source scientific papers and the generated questions, ensuring strict verification of knowledge accuracy and appropriateness as evaluation items. They assess whether questions faithfully reflect the paper's content while testing meaningful understanding. This dual-verification approach—checking both scientific correctness and QA suitability—ensures our dataset maintains the highest standards while capturing genuinely challenging, interconnected knowledge that reflects current advances in the field.

Through this rigorous process, we construct $Q_{\text{corpus}} = \{q_1, q_2, ..., q_n\}$ with over 10k expert-validated questions that form the foundation for meaningful graph construction.

## 3.2 GRAPH CONSTRUCTION

The core innovation of our approach lies in recovering the latent knowledge structure from the flat question collection. We accomplish this through rigorous graph-based methods that transform $Q_{\text{corpus}}$ into a meaningful evaluation graph. We construct a weighted undirected graph $G = (V, E, w)$ where the structure emerges naturally from the semantic relationships within our question corpus. The node set $V$ corresponds directly to our questions, with $V = Q_{\text{corpus}}$ and $|V| = n$, establishing a one-to-one mapping between graph nodes and evaluation questions.

We first employ an LLM to extract key concepts and entities from each question, generating keyword sets that capture the essential knowledge elements. This extraction process identifies domain-specific terms, scientific concepts, and their semantic roles within the question context. Then we define a hybrid similarity function $\text{sim} : V \times V \to [0, 1]$ that captures both semantic and knowledge-based relationships between questions through a multi-component approach. For questions $v_i$ and $v_j$ with respective keyword sets $K_i$ and $K_j$ extracted by the LLM, we embed all keywords and compute the similarity matrix $S_{ij} \in \mathbb{R}^{|K_i| \times |K_j|}$ where $S_{ij}[p, q] = \cos(\text{emb}(k_p^i), \text{emb}(k_q^j))$. We then compute two complementary similarity components:

**Bidirectional Matching:** Measures mutual coverage between keyword sets. We first define directional similarities:

$$\text{sim}_{i \to j} = \frac{1}{|K_i|} \sum_{p=1}^{|K_i|} \max_q S_{ij}[p, q] \tag{1}$$

$$\text{sim}_{j \to i} = \frac{1}{|K_j|} \sum_{q=1}^{|K_j|} \max_p S_{ij}[p, q] \tag{2}$$

where $\text{sim}_{i \to j}$ measures how well keywords in $K_i$ are covered by $K_j$, and vice versa. The bidirectional similarity is:

$$\text{sim}_{\text{bi}}(v_i, v_j) = \frac{1}{2} \left( \text{sim}_{i \to j} + \text{sim}_{j \to i} \right) \tag{3}$$

**Core Similarity:** Focuses on the strongest connections by selecting top-$k$ matches:

$$\text{sim}_{\text{core}}(v_i, v_j) = \frac{1}{k} \sum_{l=1}^{k} S_{ij}^{(l)} \tag{4}$$

where $S_{ij}^{(l)}$ represents the $l$-th largest element in the flattened similarity matrix, and $k = \min(\kappa, |K_i|, |K_j|)$ with $\kappa$ being a predefined parameter.

The final similarity score combines these two components:

$$\text{sim}(v_i, v_j) = \gamma \cdot \text{sim}_{\text{core}}(v_i, v_j) + (1 - \gamma) \cdot \text{sim}_{\text{bi}}(v_i, v_j) \tag{5}$$

where $\gamma \in [0, 1]$ controls the balance between core similarity and bidirectional matching.

An edge $(v_i, v_j)$ exists in our graph if and only if $\text{sim}(v_i, v_j) > \theta$, where $\theta$ is an adaptive threshold determined. Each edge carries a weight $w(v_i, v_j) = \text{sim}(v_i, v_j)$, encoding the strength of the relationship between questions. This construction ensures that our graph captures meaningful semantic relationships while avoiding noise from spurious connections.

## 4 GRAPH-BASED MODEL EVALUATION

The graph structure of our benchmark enables fundamentally new approaches to model evaluation. We leverage the evaluation graph topology to assess two core aspects of model performance—its global coherence in understanding life sciences as an integrated knowledge domain and the distribution of its errors across the graph—and develop complementary methods to address these challenges.

### 4.1 GLOBAL COHERENCE SCORE (GCS)

Consider two models achieving identical 75% accuracy on a life sciences benchmark. Model A correctly answers questions about protein synthesis, translation, and ribosome function as a coherent cluster, while Model B's correct answers scatter randomly—answering about ATP synthesis while missing basic cellular respiration, correctly identifying drug mechanisms while failing on the underlying biochemistry. Traditional accuracy metrics see these models as equivalent, yet any domain expert would immediately recognize Model A's superior understanding. The Global Coherence Score (GCS) captures this phenomenon by recognizing that genuine comprehension creates clusters of consistent performance in the knowledge graph.

For model $M$ evaluated on graph $G = (V, E, w)$, let $\text{Res}_M(v) \in \{0, 1\}$ denote whether $M$ correctly answers question $v$. We compute the neighborhood coherence for each node:

$$\text{Coherence}_M(v) = \frac{\sum_{u \in N(v)} w(v, u) \cdot \text{Res}_M(u)}{\sum_{u \in N(v)} w(v, u)} \tag{6}$$

where $N(v)$ denotes the set of nodes whose distance from $v$ is at most 1. This represents the weighted accuracy within $v$'s semantic neighborhood.

The GCS transforms each binary correct/incorrect outcome into a continuous value that reflects neighborhood support:

$$\text{GCS}(M) = \frac{1}{|V|} \sum_{v \in V} \text{Res}_M(v) \cdot \text{Coherence}_M(v) \quad (7)$$

Although the GCS is strongly correlated with overall accuracy, transforming each binary 0/1 outcome into a weighted value between 0 and 1 allows it to reveal knowledge gaps even among correctly answered nodes with low neighborhood support. This highlights cases where the model may have guessed correctly in isolation, without demonstrating genuine understanding of the surrounding semantic context—making GCS a more diagnostic measure of structured knowledge.

## 4.2 Knowledge Balance Score (KBS)

While GCS measures overall coherence, understanding how consistently a model maintains coherence across different regions of the knowledge graph reveals deeper insights into its knowledge organization patterns. Traditional evaluation metrics fail to capture whether a model exhibits balanced understanding or demonstrates highly variable performance across semantic neighborhoods. We introduce the Knowledge Balance Score (KBS), a novel metric that quantifies the variance in local coherence patterns to assess knowledge stability and balance.

For a given model $M$ evaluated on graph $G = (V, E, w)$, we leverage the neighborhood coherence values $\text{Coherence}_M(v)$ computed in the GCS framework. The KBS is computed as the variance of amplified coherence values across all nodes:

$$\text{KBS}(M) = \text{Var}(\text{Coherence}_M(v) \times \alpha) \quad (8)$$

$$= \frac{1}{|V|} \sum_{v \in V} (\text{Coherence}_M(v) \times \alpha - \mu)^2 \quad (9)$$

where $\alpha > 1$ is an amplification factor that enhances the distinction between high and low coherence regions, and $\mu = \frac{1}{|V|} \sum_{v \in V} (\text{Coherence}_M(v) \times \alpha)$ is the mean amplified coherence across all nodes.

This variance-based perspective on coherence analysis enables precise identification of whether models suffer from knowledge imbalance or systematic understanding deficits, guiding more effective training and improvement strategies.

## 5 Experiments

### 5.1 Experimental Setup

**Benchmark Details.** Our dataset construction pipeline generated high-quality questions from recent peer-reviewed literature spanning three major life sciences domains. For each subdomain, the corresponding graph is a subgraph of the global evaluation graph. Table 1 presents the structural characteristics of our constructed evaluation graph. More details can be found in Appendix A.1.

Table 1: LG-Bench: Graph-Structured Benchmark Statistics

| Domain | Questions (Nodes) | Semantic Edges | Avg. Degree |
|---|---|---|---|
| Biology | 3,630 | 521,890 | 287.50 |
| Chemistry | 3,360 | 318,763 | 189.74 |
| Medicine | 3,210 | 437,442 | 272.55 |
| **Overall** | **10,200** | **2,990,277** | **586.32** |

**Models.** We evaluate a comprehensive set of large language models spanning different scales, architectures, and specializations to provide a thorough assessment of capabilities. In the case of

open-source models , we include the Qwen2.5 series (0.5B/7B/14B/32B/72B) (Team, 2024), the Llama family (Touvron et al., 2023; Meta AI, 2024; Dubey et al., 2024) including Llama-7B, Llama-3-8b, Llama-3.3-70B, Llama-3.1-405B, and Llama-4-scout. We also evaluate the GLM-4 series (9B/32B) (GLM et al., 2024), Gemma-3-1B (Team, 2025), and DeepSeek-v3 (DeepSeek-AI, 2024). For domain-specific models, we include BioMedLM-2.7B (Bolton et al., 2024), Medicine-LLM-7B (Cheng et al., 2024), HuatuoGPT family (Zhang et al., 2023; Chen et al., 2023), and Llama3-Med42-8B (Christophe et al., 2024), which are specifically fine-tuned for life sciences applications.

For closed-source commercial models, we utilize the GPT series including GPT-3.5, GPT-4o-mini, GPT-4o, OpenAI o3, and OpenAI o4-mini (OpenAI, 2023a;b), the Claude family with Claude-3.5-sonnet and Claude-3.7-sonnet (Anthropic, 2024; 2025), and the Grok series (Grok-2/Grok-3) (xAI, 2025). All models are evaluated using identical prompting strategies and evaluation protocols to ensure fair comparison.

## 5.2  RESULTS

Table 2 presents comprehensive evaluation results across all models on LG-Bench using our graph-based metrics. All accuracy (Acc) and Global Coherence Score (GCS) values are reported as percentages. For KBS computation, we set the amplification parameter $\alpha = 100$ to enhance the distinction between high and low coherence regions. All experiments were conducted three times and averaged, with more experimental settings available in Appendix B. Our evaluation demonstrates significant performance variations across models, with overall accuracy ranging from 23.93% (BioMedLM-2.7B) to 89.51% (OpenAI o4-mini), GCS spanning 5.67% to 80.59%, and KBS values varying from 5.28 to 27.60. Notably, our graph-based metrics reveal substantial differences between models that are not captured by traditional accuracy alone, effectively distinguishing models with coherent knowledge organization from those exhibiting scattered understanding patterns across the life sciences domain. As shown in Fig.4, we compare the responses of Qwen2.5 within a local graph structure and observe a clear lack of understanding of certain concepts in Qwen2.5-7B, whereas Qwen2.5-72B demonstrates a clear and solid grasp of these knowledge points. A more detailed analysis is provided in Appendix A.2.

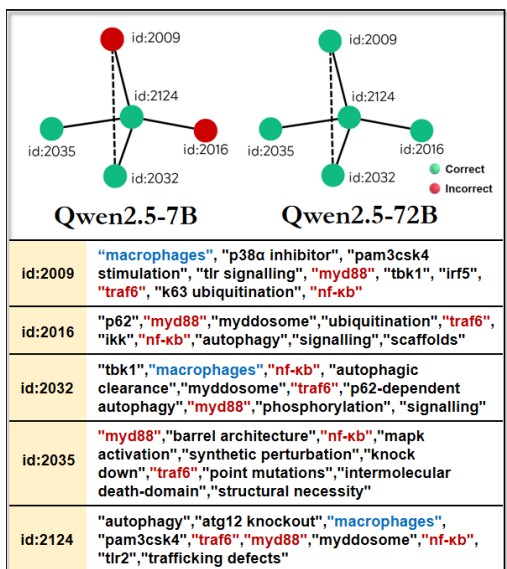

Figure 4: Qwen2.5 results within a local graph structure. Qwen2.5-7B failed on many questions in the neighborhood-level evaluation. Red keywords indicate areas of severe knowledge weakness, while blue keywords denote relatively weak points. In contrast, Qwen2.5-72B correctly answered all relevant questions in the neighborhood test, demonstrating a relatively good grasp of the associated knowledge.

**Open-Weight LLMs *vs.* Closed-Weight LLMs.** The result highlights a striking contrast between open and closed models when confronted with LG-Bench, whose items deliberately target clinical guidelines, drugs, and biomolecular findings released recently. Early open models such as HuatuoGPT, BioMedLM, and Llama-7B reach only 24–31% accuracy, revealing substantial blind spots for the newest knowledge. In contrast, iterative open-source versions like HuatuoGPT2-7B, Llama-3-8B, and Llama3-Med42-8B boost the same metric to 60–78% while staying under 10B parameters. These improvements attest to the compounding benefits of an open ecosystem, where community-contributed data refreshes and lightweight domain fine-tuning shorten the model–data–task loop. Closed-source systems still occupy the top tier: OpenAI o3 and OpenAI o4-mini surpass 88% overall accuracy, reflecting the value of proprietary corpora and extensive RLHF (Christiano et al., 2017). Nevertheless, the gap is narrowing; fully open models like Qwen2.5-72B and DeepSeek-v3 trail GPT series by less than 3 pp. In sum, while closed models presently set the performance ceil-

Table 2: Evaluation Results on LG-Bench: Acc denotes traditional accuracy (in %), GCS measures global coherence (in %), with higher scores indicating stronger capabilities. We use **bold** to highlight the best-performing model in each domain, and *italics* to indicate the second-best. KBS quantifies knowledge balance, and higher scores indicate a more uneven distribution.

| Model | Medicine | | | Biology | | | Chemistry | | | Overall | | |
|---|---|---|---|---|---|---|---|---|---|---|---|---|
| | Acc | GCS | KBS | Acc | GCS | KBS | Acc | GCS | KBS | Acc | GCS | KBS |
| Open-Weight LLM (Scale<10B) | | | | | | | | | | | | |
| Qwen2.5-0.5B | 37.07 | 13.96 | 22.19 | 38.10 | 14.83 | 26.34 | 37.44 | 14.35 | 36.52 | 37.56 | 14.39 | 16.11 |
| Gemma-3-1b | 51.56 | 26.73 | 29.42 | 55.45 | 30.71 | 21.97 | 52.65 | 28.20 | 39.34 | 53.30 | 28.62 | 17.94 |
| BioMedLM-2.7B | 24.14 | 5.80 | 12.85 | 23.80 | 5.65 | 9.03 | 23.87 | 5.56 | 13.69 | 23.93 | 5.67 | 5.28 |
| Llama-7B | 30.06 | 9.19 | 13.62 | 30.91 | 9.60 | 14.41 | 32.20 | 10.47 | 19.14 | 31.07 | 9.76 | 6.80 |
| Medicine-LLM-7B | 38.72 | 15.23 | 22.23 | 34.88 | 12.26 | 17.23 | 35.57 | 12.58 | 23.87 | 36.31 | 13.29 | 10.65 |
| HuatuoGPT-7B | 28.75 | 8.36 | 15.94 | 29.20 | 8.50 | 10.31 | 28.42 | 8.13 | 12.97 | 28.80 | 8.33 | 5.96 |
| HuatuoGPT2-7B | 61.06 | 38.09 | 33.85 | 62.42 | 39.36 | 20.89 | 58.36 | 34.41 | 44.45 | 60.66 | 37.39 | 20.00 |
| Qwen2.5-7B | 78.38 | 62.15 | 29.17 | 80.03 | 64.24 | 21.58 | 76.76 | 59.21 | 25.74 | 78.43 | 62.00 | 15.19 |
| Llama-3-8b | 72.77 | 54.11 | 35.65 | 75.67 | 57.64 | 17.94 | 72.53 | 53.16 | 27.97 | 73.73 | 55.04 | 14.62 |
| Llama3-Med42-8B | 78.10 | 61.60 | 25.61 | 77.38 | 60.01 | 18.22 | 75.00 | 56.71 | 30.78 | 76.82 | 59.43 | 13.56 |
| GLM-4-9b | 76.67 | 59.61 | 30.59 | 77.47 | 59.82 | 25.99 | 75.45 | 57.50 | 36.28 | 76.55 | 58.97 | 18.05 |
| Open-Weight LLM (Scale>10B) | | | | | | | | | | | | |
| Qwen2.5-14B | 83.02 | 69.62 | 20.67 | 85.26 | 72.72 | 18.54 | 82.62 | 68.64 | 21.91 | 83.69 | 70.42 | 11.73 |
| Qwen2.5-32B | 84.70 | 72.55 | 21.01 | 86.06 | 74.34 | 15.02 | 83.75 | 70.38 | 20.78 | 84.87 | 72.56 | 11.07 |
| GLM-4-32B | 78.94 | 63.32 | 29.30 | 80.44 | 64.82 | 22.22 | 78.30 | 61.77 | 29.31 | 79.26 | 63.33 | 15.28 |
| Llama-3.1-70B | 72.77 | 69.51 | 24.30 | 83.14 | 69.19 | 19.55 | 71.67 | 66.78 | 24.67 | 82.54 | 68.58 | 12.70 |
| Qwen2.5-72B | 86.26 | 75.12 | 16.53 | 86.97 | 75.68 | 15.45 | 84.85 | 72.44 | 23.04 | 86.05 | 74.52 | 10.26 |
| Llama-3.1-405b | 88.07 | 78.13 | 13.98 | 88.26 | 78.01 | 10.88 | 87.14 | 76.18 | 14.98 | 87.83 | 77.51 | 7.19 |
| Llama-4-scout | 78.69 | 62.61 | 32.32 | 81.79 | 66.95 | 25.66 | 80.27 | 64.79 | 28.76 | 80.31 | 64.89 | 18.65 |
| Deepseek-v3 | 87.17 | 76.57 | 14.76 | 87.96 | 77.51 | 11.63 | 86.90 | 75.70 | 15.09 | 87.36 | 76.70 | 7.50 |
| Closed-Weight LLM | | | | | | | | | | | | |
| Grok-2 | 88.54 | 78.71 | 16.61 | 89.01 | 79.30 | 11.15 | 87.23 | 76.36 | 17.60 | 88.27 | 78.20 | 7.78 |
| Grok-3 | 86.29 | 75.06 | 14.01 | 84.38 | 71.75 | 29.89 | 85.77 | 73.75 | 16.68 | 85.44 | 73.54 | 10.40 |
| Claude-3.5-sonnet | 88.91 | 79.43 | 12.51 | 88.65 | 78.61 | 10.25 | 86.93 | 76.03 | 18.69 | 88.17 | 78.09 | 7.56 |
| Claude-3.7-sonnet | 88.41 | 78.76 | 15.91 | 89.04 | 79.29 | 9.00 | 87.83 | 77.27 | 14.71 | 88.44 | 78.52 | 6.97 |
| GPT-3.5 | 76.42 | 59.35 | 27.79 | 77.02 | 59.67 | 23.01 | 74.55 | 56.38 | 35.35 | 76.02 | 58.47 | 16.31 |
| GPT-4o-mini | 82.21 | 68.77 | 25.94 | 84.41 | 71.13 | 14.91 | 81.70 | 67.26 | 26.46 | 82.82 | 69.19 | 12.68 |
| GPT-4o | 88.47 | 78.99 | 15.08 | *89.70* | *80.44* | 10.62 | 88.01 | 77.79 | 13.29 | 88.75 | 79.19 | 6.22 |
| OpenAI o3 | **90.00** | **81.57** | 10.65 | 89.26 | 79.84 | 9.94 | **89.05** | **79.56** | 11.60 | *89.42* | *80.34* | 5.83 |
| OpenAI o4-mini | *89.75* | *81.09* | 10.69 | **89.89** | **81.14** | 7.22 | *88.87* | *79.32* | 9.46 | **89.51** | **80.59** | 4.45 |

ing, rapid, community-driven iteration is continuously raising the open-source floor, accelerating progress for medical reasoning at large.

**Model Scale Effects.** Our empirical analysis confirms a clear parameter–performance scaling law. Sub-10B models deliver only moderate results, with accuracies below 80% and GCS under 62%; for instance, Qwen2.5-0.5B attains merely 37.56% accuracy and a GCS of 14.39%. Once the parameter count exceeds 10B, every metric improves sharply. Within the Qwen2.5 family, accuracy and GCS rise from 83.69% and 70.42% at 14B to 84.87% and 72.56% at 32B, and further to 86.05% and 74.52% at 72B. Knowledge balance benefits from scaling as well: the KBS drops monotonically from 16.11 at 0.5B to 11.73 at 14B, 11.07 at 32B, and 10.26 at 72B, indicating progressively more coherent knowledge organization. The Llama series exhibits the same scaling advantages, under-scoring the generality of these trends across model families. The Coherence Score distrubution of Qwen2.5 family is shown as Fig. 5.

**Impact of Domain-Specific Training.** Our graph-based metrics provide a detailed lens through which to analyze the effects of domain-specific fine-tuning. A direct comparison between the generalist Llama-3-8b and its domain-adapted counterpart, Llama3-Med42-8B, reveals that specialization does more than just increase accuracy—it fundamentally reshapes the model's knowledge structure. We embed each knowledge graph with *node2vec* (Grover & Leskovec, 2016) into a 2-D manifold and plot Coherence Score heatmaps, shown as Fig. 6. Although the overall surface becomes brighter—corroborating the aggregate metric improvements—an inverted pattern emerges in certain regions, revealing a migration of knowledge density rather than a uniform amplification.

In the target domain of Medicine, Llama3-Med42-8B shows a substantial improvement in its GCS, rising to 61.60% from the base model's 54.11%. More revealing is the impact on the KBS. The KBS

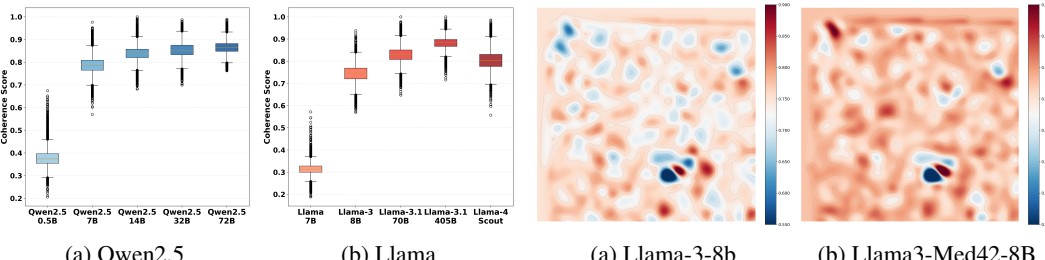

<table>
<tr><td>(a) Qwen2.5</td><td>(b) Llama</td><td>(a) Llama-3-8b</td><td>(b) Llama3-Med42-8B</td></tr>
</table>

Figure 5: Coherence Score distribution of Qwen2.5 and Llama family. As model size increases, its overall capability in the life sciences domain gradually improves, while its knowledge becomes more evenly distributed.

Figure 6: Heatmap visualization of the LG-Bench, where each cell represents a node's Coherence Score. Redder regions indicate higher coherence, while bluer areas indicate lower coherence.

for Llama3-Med42-8B in Medicine drops sharply to 25.61 from the base model's 35.65. This significant reduction in variance demonstrates that the fine-tuning not only enhanced knowledge but also homogenized it, leading to a more evenly distributed and balanced understanding across different medical topics. However, this specialization comes with trade-offs that metrics can precisely identify. While its GCS shows minor improvements in the related field of Biology, Llama3-Med42-8B's KBS in Chemistry increases to 30.78, higher than the base model's 27.97. This suggests that the intense focus on medicine may have inadvertently created a more imbalanced and "spiky" knowledge representation in the less related chemical domain. This demonstrates that domain adaptation can reshape knowledge organization both positively and negatively across different semantic regions, a critical insight that flat accuracy metrics would completely miss.

**Domain-Specific Performance Patterns.** A discipline-level decomposition reveals pronounced performance asymmetries, as illustrated in Fig. 7. Biology most often emerges as the relative strong models such as GPT-4o-mini, Qwen2.5-72B, Claude-3.7, HuatuoGPT2-7B achieve their peak accuracies and GCS in this domain, yet this advantage is not universal, varying with architecture and scale. Chemistry, by contrast, remains the chief bottleneck: even state-of-the-art systems record their lowest scores there, a deficit plausibly tied to the field's dense symbolic notation, heterogeneous nomenclature, and high conceptual abstraction. Such disciplinary disparities highlight the necessity of domain-specific evaluation frameworks that can capture the unique challenges and knowledge structures inherent to each field.

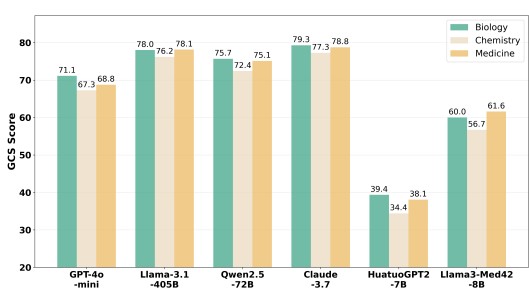

Figure 7: GCS Comparison Across Models and Categories.

# 6 CONCLUSION AND OTHERS

**Conclusion.** In this work, we introduced LG-Bench, the first graph-structured benchmark designed to address the fundamental limitations of traditional evaluations in the life sciences. By modeling knowledge as an interconnected graph and introducing novel coherence-based metrics, we provide the community with the tools to move beyond simple accuracy and assess the depth of a model's scientific reasoning.

**Future Work.** We will continuously update our dataset and incorporate semantic information into the edges of the graph structure to enable more fine-grained evaluation.

**Broad Impact.** We believe this rigorous evaluation framework will guide LLMs beyond pattern-matching toward becoming true scientific partners. Ultimately, more reliable and coherent AI systems will accelerate discovery and innovation across the life sciences.

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

# A  APPENDIX

## A.1  DETAILS OF LG-BENCH

### A.1.1  HYPERPARAMETER SETTINGS

Each edge in our evaluation graph represents a weighted semantic relationship between questions, computed using our bidirectional matching and core similarity algorithms. For graph construction, we set the balance parameter $\gamma = 0.6$ to weight core similarity slightly higher than bidirectional matching, selected the top $\kappa = 3$ strongest connections for core similarity computation, and applied an adaptive edge threshold $\theta = 0.4$ to ensure meaningful semantic relationships while maintaining graph connectivity.

### A.1.2  SAMPLE QUESTIONS AND GRAPH STRUCTURE FROM LG-BENCH

**Sample Questions.** The following section presents several representative examples from our dataset, including options and explanations. Correct answers are highlighted in red.

---

**Question 48:** What primary cellular phenotype links loss of keratinization genes to increased EAC aggressiveness?

- (A) Suppression of Wnt signaling curtailing stem-cell renewal
- **(B) Enhanced epithelial plasticity facilitating invasion**
- (C) Elevated oxidative respiration reducing hypoxia tolerance
- (D) Activation of DNA damage checkpoints halting proliferation

**Explanation:** Loss of terminal squamous differentiation increases plasticity, enabling invasion and aggressive tumor behavior.

---

**Question 915:** If imaging-only maps recover large assemblies better than small ones, what physical explanation best accounts for this?

- (A) Small complexes have lower protein copy numbers, depleting peptide coverage in MS.
- **(B) Spatial resolution of light microscopy limits detection of nanometre-scale complexes, causing small assemblies to be visually indistinguishable.**
- (C) Image segmentation algorithms preferentially crop large structures for analysis.

- (D) Large assemblies generate brighter fluorescence signal enabling better antibody binding.

**Explanation:** Optical diffraction restricts resolution ( 200 nm); small complexes below this scale cannot be visually separated, whereas larger assemblies occupy resolvable regions.

---

**Question 1767:** When donor-to-recipient cell ratios in MitoTRACER coculture increased from 1:10 to 1:1, the percentage of green-converted cancer cells rose. This demonstrates that mitochondrial transfer probability depends mainly on which factor?

- **(A) Donor cell abundance creating more contact opportunities**
- (B) Recipient cell cell-cycle phase
- (C) Mitochondrial fission rate in recipients
- (D) Serum glucose concentration

**Explanation:** Higher donor proportion increases physical interactions and nanotube formation, elevating organelle transfer frequency.

---

**Question 2148:** What experimental evidence argues that myddosome clearance is p62-dependent?

- (A) p62 overexpression reduced IL-6 secretion
- (B) ProteoStat staining increased in p62-deficient cells
- **(C) p62 knockout macrophages accumulated residual myddosome structures long after signaling subsided**
- (D) Phospho-p65 levels declined faster in p62 knockout cells

**Explanation:** Persistence of remnants specifically in p62-null cells indicates its necessity for targeting complexes to autophagy.

---

**Question 3992:** Stability studies demonstrated that folate-diketone remained bound to CovCAR at pH 4.5, whereas folate-FITC dissociated. This finding predicts which therapeutic advantage?

- (A) Faster renal clearance of the adapter
- **(B) Enhanced trafficking through acidic tumour microenvironments**
- (C) Reduced requirement for lymphodepletion
- (D) Lower likelihood of cytokine release syndrome

**Explanation:** Retention of covalent linkage under acidic conditions supports sustained CAR-adapter association in endosomes and acidic tumour niches.

---

**Question 4134:** Circular dichroism of the phenanthroline chromophore shows a negative Cotton effect at 330 nm for (+)-5-[Cu][Lu]. What stereochemical element is most directly inferred from this observation?

- (A) $\Delta$-helicity at the Lu(III) helicate
- **(B) $\Lambda$-helicity at the Cu(I)-dpp clasp**
- (C) Presence of racemic knot mixture
- (D) Metal-free macrocycle formation

**Explanation:** A negative Cotton effect for bis-dpp Cu(I) complexes corresponds to Λ helicity, indicating that the clasp crossing in the knot adopts Λ configuration.

---

**Question 7441:** To evaluate whether N-0385 can block influenza A virus entry in vitro, which cell model would best replicate the TMPRSS2-dependent activation step targeted in the Nature study, and why?

- **(A) Human Calu-3 airway epithelial cells, because they endogenously express TM-PRSS2 and other TTSPs needed for viral fusion**
- (B) BHK-21 hamster fibroblasts, because they lack serine proteases that interfere with fusion assays
- (C) Vero E6 kidney cells, because they are routinely used for high-titre virus propagation despite minimal TMPRSS2 expression
- (D) HEK-293 human embryonic kidney cells, because they overexpress ACE2 after transient transfection

**Explanation:** Calu-3 cells were used for their endogenous TMPRSS2 activity, allowing peptidomimetic inhibitors to block spike/hemagglutinin activation.

---

**Question 8622:** Vorinostat targets class I/II HDACs. Which downstream effect most directly increases tumor immunogenicity?

- (A) Blocking VEGF secretion, thereby reducing angiogenesis.
- **(B) Enhanced histone acetylation leading to up-regulation of MHC class I genes.**
- (C) Suppression of DNA repair enzymes, causing mutational overload.
- (D) Direct phosphorylation of STAT3, activating immune checkpoints.

**Explanation:** HDAC inhibition acetylates chromatin and increases expression of antigen-presentation molecules, improving immune recognition.

---

**Question 9437:** Which control would MOST convincingly demonstrate that the loss of stress-granule assembly after RIOK1 shRNA is on-target?

- (A) Overexpress GFP to control for lentiviral transduction.
- **(B) Re-express an shRNA-resistant RIOK1 cDNA and test whether SG formation is rescued.**
- (C) Add cycloheximide to dissolve granules in all samples.
- (D) Include a non-targeting shRNA vector in parallel cultures.

**Explanation:** Functional rescue with an shRNA-proof construct specifically attributes the phenotype to RIOK1 depletion.

---

**Question 9885:** A researcher cultures iPSC-derived cardiac fibroblasts on 2 kPa hydrogels and treats them with exogenous TGF$\beta$. What combined manipulation would most effectively restore quiescence according to the study's findings?

- (A) Blockade of IL-1$\beta$ signalling with anakinra alone
- (B) Knockdown of YAP together with ROCK inhibition
- (C) Overexpression of SORBS2 together with blebbistatin

- **(D) Addition of SB431542 together with saracatinib**

**Explanation:** Soft substrate plus TGF$\beta$ inhibition alone was insufficient after activation; adding SRC inhibition (saracatinib) created the synergy required for full reversal toward quiescence.

---

**Sample Graph Structure.** We sample several local subgraph structures and visualize them, as shown in Fig. 8. We also present the neighboring information, as shown in Table 3. It can be observed that the graph structure of our dataset captures rich semantic connections.

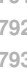

Figure 8: Visualization of a subset of LG-Bench, where red nodes represent sampled questions and their top-5 semantic neighbors (orange) based on edge weights. All other nodes and edges are displayed in gray to show overall connectivity. The blue edges indicate the strongest semantic connections (top-5 weighted edges) for each sampled node.

### A.2 DETAILS OF EVALUATION

#### A.2.1 EVALUATION ENVIRONMENT

For models with open weights, we downloaded the official weights and deployed tests on a cluster with 8 nodes as shown in the Table 4. We set the same default temperature parameters and tested with the exact same prompt. For models with closed weights, we conducted tests based on the API while maintaining the same testing configurations as the open-weight models to ensure fairness.

#### A.2.2 CASE STUDY

To further analyse the performance differences between pre-trained language models of varying sizes in complex question-answering tasks, we selected a set of highly relevant questions as evaluation samples to compare the response quality of large models (Qwen-72B) and medium-sized models (7B) under the same input settings. We retained the original question stems, concealed the multiple-choice options, and prompted the models to perform open-ended question answering. Qwen-72B can answer this set of questions, consistently generating responses with clear structural logic, accurate content alignment, and a deep understanding of domain-specific mechanisms. In contrast,

Table 3: Top-5 highest-weight neighbors of 20 randomly sampled nodes. Weight (w) indicates semantic similarity.

| Node ID | Degree | Neighbor 1 (w) | Neighbor 2 (w) | Neighbor 3 (w) | Neighbor 4 (w) | Neighbor 5 (w) |
|---------|--------|----------------|----------------|----------------|----------------|----------------|
| 48 | 191 | 39 (0.6531) | 465 (0.6498) | 9952 (0.6269) | 497 (0.6089) | 29 (0.5966) |
| 4552 | 239 | 4564 (0.6253) | 4538 (0.6165) | 4524 (0.6103) | 4587 (0.5998) | 8973 (0.5913) |
| 7441 | 204 | 7448 (0.8288) | 7690 (0.7892) | 2851 (0.7648) | 7685 (0.7440) | 7445 (0.7353) |
| 2822 | 471 | 2824 (0.8452) | 2828 (0.8393) | 3419 (0.8200) | 2829 (0.7956) | 3416 (0.7751) |
| 915 | 478 | 2446 (0.7431) | 10071 (0.7266) | 6640 (0.6843) | 8843 (0.6815) | 3149 (0.6744) |
| 9216 | 61 | 9086 (0.6868) | 9078 (0.6770) | 9061 (0.6594) | 9101 (0.6590) | 9211 (0.6177) |
| 9885 | 100 | 9735 (0.7791) | 9883 (0.7493) | 9881 (0.7485) | 9732 (0.7007) | 9739 (0.6861) |
| 9437 | 195 | 9032 (0.7917) | 9045 (0.7405) | 9022 (0.7328) | 7376 (0.7022) | 9012 (0.6970) |
| 9604 | 297 | 9610 (0.8237) | 9955 (0.8006) | 9606 (0.7000) | 7740 (0.6839) | 7753 (0.6775) |
| 4134 | 66 | 4138 (0.6928) | 4171 (0.6872) | 4168 (0.6585) | 4180 (0.6016) | 4144 (0.5986) |
| 2707 | 113 | 5536 (0.7635) | 3346 (0.7352) | 5531 (0.6819) | 3341 (0.6663) | 5899 (0.6505) |
| 388 | 187 | 528 (0.8545) | 513 (0.8285) | 379 (0.8248) | 550 (0.7506) | 548 (0.7412) |
| 2729 | 104 | 2752 (0.8181) | 3320 (0.7863) | 2739 (0.7591) | 2714 (0.7580) | 2721 (0.7579) |
| 8047 | 116 | 8574 (0.7978) | 8044 (0.6790) | 8595 (0.6327) | 8588 (0.6265) | 8058 (0.6261) |
| 1767 | 484 | 2084 (0.8085) | 2066 (0.8027) | 2064 (0.7842) | 1785 (0.7794) | 1795 (0.7515) |
| 8622 | 598 | 8616 (0.8116) | 8062 (0.7839) | 8086 (0.7676) | 8614 (0.7465) | 8633 (0.7454) |
| 9528 | 314 | 7740 (0.8151) | 7734 (0.7978) | 9410 (0.7548) | 7739 (0.6999) | 8164 (0.6966) |
| 758 | 84 | 784 (0.7270) | 753 (0.6624) | 776 (0.6561) | 783 (0.6472) | 760 (0.6453) |
| 2148 | 587 | 2108 (0.8436) | 2016 (0.8381) | 2001 (0.8345) | 2131 (0.8325) | 2129 (0.8256) |
| 3992 | 169 | 3983 (0.7378) | 3996 (0.7156) | 807 (0.6854) | 4007 (0.6290) | 4005 (0.6283) |

Table 4: Experimental Environment

| Component | Specification |
|-----------|---------------|
| **CPU** | |
| Model | Intel(R) Xeon(R) Platinum 8336C |
| Total Cores | 128 |
| Total Threads | 128 |
| Max Turbo Frequency | 2.30GHz |
| **GPU** | |
| Model | NVIDIA A800-SXM4 $\times 8$ |
| VRAM | 80 GB GDDR6X |

while the 7B model's responses demonstrate some structural and organisational coherence, they still exhibit errors in certain questions, with significantly insufficient depth of understanding, expression precision, and knowledge mobilisation capabilities. For example, in the id2016 question, Qwen-7B overlooks the core mechanism of "ubiquitin-dependent transfer to autophagy" and instead provides a vague "autophagy-related" explanation. Additionally, the 7B model disregarded the role of the "MyD88-dependent myddosome complex" and erroneously assumed it influences early "NF-$\kappa$B-dependent transcription".

This misunderstanding recurs in similar questions or adjacent nodes, indicating the model struggles to effectively reuse contextual information and lacks a robust representation of the internal structure of pathway mechanisms. Furthermore, larger-scale models demonstrate stronger performance in contextual understanding and domain knowledge integration, while smaller-scale models often misidentify key mechanisms. This may stem from their insufficient understanding of domain-specific signalling pathways and their inability to distinguish between mechanistically similar components. Despite the five samples sharing high-frequency domain-specific keywords—such as "MyD88", "traf6", and "nf-$\kappa$B",the smaller models (Qwen2.5-7B) still cannot consistently generalise mechanisms across samples. This inconsistency manifests not only as factual errors but also as failures in mechanism inference-for instance, missing prior knowledge of the"ubiquitin-mediated autophagy pathway" in related cases (id2032 and id2035). As the number of parameters expanded from 7B to 72B, the Qwen model showed significant improvements in consistency and depth in its understanding of professional mechanisms. This finding aligns with the conclusions drawn in the main text regarding differences in knowledge organization coherence, thereby substantiating the efficacy of graph structure indicators in differentiating model capabilities.

## B    LLM Usage

A large language model (LLM) was used for drafting and language polishing of this article. Beyond these uses, no AI tools were involved in study design, core experiments, result analysis, or interpretation. The authors are fully responsible for the accuracy and integrity of the work.

