# OpenReview forum: "LG-Bench: A Graph-Structured Evaluation Benchmark for Life Sciences"
_ICLR.cc/2026/Conference — Submitted to ICLR 2026_

### Official Review · Reviewer_f3N5 · 2025-10-20

**Soundness:** 3
**Presentation:** 3
**Contribution:** 3
**Rating:** 6
**Confidence:** 4

**Summary:**

This paper introduces LG-Bench, that contains over 10,000 high quality multi-choice questions capturing topoplogical structure of knowledge in the domain of biology, chemistry and medicine. The authors further introduce GCS and KBS to track performance across knoweldge clusters, providing deeper insights of performance on the knowledge distribution rather than plainly reporting accuracy by treating each individual questions as flat and independent entities. Evaluation and Results demonstrate that large-sized closed-sourced models perform quite well in preserving knowlegde within a single and across different domains. Further insights reveal that fine-tuning alters the knowledge landscape by re-adjusting the weights.

**Strengths:**

- The paper is well written with proper flow and clear/readable diagrams.
- Experiments conducted are well-rounded and covered all bases. Interesting plots have been demonstrated highlighting the key impact of the evaluation metric. Overall the need for the proposed evaluation metric is motivated well.
- The convolution of evaluation over a knowledge graph is novel and can inspire future research to gauge models on different levels of topology.

**Weaknesses:**

- The weighted accuracy w(v,u) in line 269 has not been explained properly.
- The motivation behind the design choices made towards stage-2 of the benchmark construction pipeline (Graph construction) is not provided. Were there other keyword representation you tried? How does the breakdown and computing semantic similarity over the keyword beneficial? How many keywords were extracted per question and what was their granularity? Moreover, what was the reason for choosing a weighted combination of bi-directional similarity and core-similarity? It would be great if you can demonstrate some ablation studies to justify the design choices made.

**Questions:**

1) What speciaized LLM was used for Guided Question Generation (line 154)?
2) How were the hyper-parameters decided? Was it a theoritical choice or empirically based?
3) Can you share the prompts used for Guided Question Generation and Multi-Model Committee Assessment?
4) How many questions were passed on from Multi-Model Assessment to expert human verification? What was the portion of questions were retained from the committee check and what portion was introduce after expert verification?

---

> ### Author Response · Authors · 2025-11-14
> **Response to Reviewer f3N5**
>
> We sincerely thank you for your thorough and constructive review. We are greatly encouraged by your positive assessment of our work, particularly your recognition of the novelty of our graph-based evaluation approach, the quality of our presentation, and the comprehensiveness of our experiments. Your acknowledgment that our method "can inspire future research to gauge models on different levels of topology" validates our core contribution.
>
> We appreciate your detailed questions and suggestions, which will help us strengthen our paper. Below, we address each of your concerns systematically, providing additional clarifications and implementation details.
>
> Responses to Questions:
>
> Q1: What speciaized LLM was used for Guided Question Generation (line 154)?
>
> A1: We use Claude-Opus-4 for Guided Question Generation (line 154) due to its exceptional reasoning and generation capabilities. At the same time, we employ GPT-4o to conduct evidence grounding.
>
> Q2: How were the hyper-parameters decided? Was it a theoritical choice or empirically based?
>
> A2: It was selected based on empirical experience. To address concerns about sensitivity, we added a series of ablations varying the hyperparameters to induce sparser graphs. Our ablation experiments are provided in the global response section at the top.
>
> Q3: Can you share the prompts used for Guided Question Generation and Multi-Model Committee Assessment?
>
> Due to space limitations, we only present a subset of the prompts here. We will release all prompts and the full generation framework.
>
> Guided Question Generation:
>
> ```txt
>     You are an expert content creator specializing in designing domain-specific, knowledge-based assessments for life sciences.
>
>     You will be provided with:
>     1. A research paper in the life sciences (e.g., biology, medicine)
>     2. A target difficulty level: {difficulty_level}
>     3. Bloom's taxonomy levels: {bloom_levels}
>     4. Difficulty description: {difficulty_description}
>
>     Your task is to generate a precise, instructional prompt that will guide another AI agent to create high-quality multiple-choice questions (MCQs) at the specified {difficulty_level}. The resulting questions must assess biologically meaningful knowledge and scientific understanding derived from the paper.
>
>     Important Global Rules:
>     1. Stand-Alone Questions: Question stems must read as self-contained science items.
>       • DO NOT use phrases like "in the paper," "according to the study," "as described by the authors," etc.
>       • Embed the scientific knowledge naturally, so questions look like a conventional test bank rather than a reading-comprehension quiz.
>
>     2. Focus Scope:
>       • Biological, biochemical, medical, or pharmacological concepts introduced or explained in the paper.
>       • Avoid trivialities (dataset names, figure numbers, author affiliations).
>
>     3. Question Types & Cognitive Skills:
>       • Knowledge-oriented: definitions, classifications, functions, relationships, mechanisms.
>       • Align with {bloom_levels} for the given {difficulty_level}.
>       • Test recognition, comprehension, or application (depending on difficulty), not rote details.
>
>     4. Distractor Quality:
>       • Scientifically plausible, relevant, and non-obvious.
>       • Leverage common misconceptions or closely related concepts to raise challenge.
>
>     5. Suggested Concept Areas to Mine from the Paper:
>       • Functional roles of genes/proteins/pathways.
>       • Mechanisms of disease or therapeutic action.
>       • Diagnostic markers, disease progression indicators.
>       • Significant experimental findings establishing cause–effect or structure–function relationships.
>       • Key concepts from molecular biology, neuroscience, pharmacology, genetics, etc.
>
>     6. Output Requirements: Do not include or request generation templates or examples.
> 		...
>
> ```
>
> Multi-Model Committee Assessment:
>
> ```txt
>     You are an expert educational assessment evaluator specializing in scientific multiple-choice questions.
>
>     Please evaluate the following multiple-choice question across four dimensions and provide scores from 0.0 to 1.0 for each:
>
>     **QUESTION TO EVALUATE:**
>     Question: {question}
>
>     Options:
>     A) {option_A}
>     B) {option_B}
>     C) {option_C}
>     D) {option_D}
>
>     Correct Answer: {correct_answer}
>     Explanation: {explanation}
>     Bloom Level: {bloom_level}
>
>     **EVALUATION CRITERIA:**
>
>     1. **Question Quality (0.0-1.0)**:
>        - Is the question clear, unambiguous, and well-structured?
>        - Does it match the stated Bloom's taxonomy level?
>        - Is the difficulty appropriate and consistent?
>     2. **Option Quality (0.0-1.0)**:
>  ...
>     3. **Paper Alignment (0.0-1.0)**:
>  ...
>     4. **Assessment Suitability (0.0-1.0)**:
>  ...
> ```

---

> > ### Author Response · Authors · 2025-11-14
> > **Response to Reviewer f3N5 (2)**
> >
> > Q4: How many questions were passed on from Multi-Model Assessment to expert human verification? What was the portion of questions were retained from the committee check and what portion was introduce after expert verification?
> >
> > A4: We first generated 13,250 questions based on life science papers. A multi-model committee then filtered out 2,653 low-quality questions (e.g., shallow, lacking textual evidence, or structurally flawed), leaving 10,597 higher-quality questions. During expert validation, we verified  4,153 questions, ultimately removing 397 questions. The final number of questions which was checked by experts is 3,756. The tables below show the details of  each stage.
> >
> > | Stage   | Description                                  | Number         |
> > | ------- | -------------------------------------------- | -------------- |
> > | Stage 1 | Initially generated question (LLM)           | 13,250         |
> > | Stage 2 | After selecting by the Multi-Model Committee | 10,597 (-2653) |
> > | Stage 3 | After experts review（final reservation）    | 10,200 (-397)  |
> >
> >
> > | Error Type                         | Number |
> > | ---------------------------------- | ------ |
> > | Logical or factual errors          | 47     |
> > | Low relevance to the domain        | 214    |
> > | Topics not suitable for evaluation | 136    |
> >
> >
> > Responses to Weakness:
> >
> > 1. The weighted accuracy w(v,u) in line 269 is defined in line 242 (similarity).
> >
> > 2. The motivation behind the design choices made towards stage-2 of the benchmark construction pipeline (Graph construction) is not provided. Were there other keyword representation you tried? How does the breakdown and computing semantic similarity over the keyword beneficial? How many keywords were extracted per question and what was their granularity? Moreover, what was the reason for choosing a weighted combination of bi-directional similarity and core-similarity? It would be great if you can demonstrate some ablation studies to justify the design choices made
> >    1. **Were there other keyword representation you tried?** Yes, we experimented with various keyword representation methods. Four domain experts manually examined keywords extracted from 400 questions and iteratively adjusted the parameters to achieve better balance between low-level (specific concepts) and high-level (thematic connections) representations.
> >
> >    2. **How does the breakdown and computing semantic similarity over the keyword beneficial?** Breaking down and computing semantic similarity over keywords allows us to eliminate redundant semantics and achieve better performance. This approach ensures that we capture meaningful relationships while avoiding overlap in semantic representations.
> >
> >    3. **How many keywords were extracted per question and what was their granularity?** We extracted 5-20 keywords per question, with granularity ranging from macro-level thematic concepts to micro-level specific details. This multi-granular approach ensures comprehensive coverage of both broad domain understanding and fine-grained scientific concepts.
> >
> >    4. **What was the reason for choosing a weighted combination of bi-directional similarity and core-similarity?** We intentionally encode both low-level semantics (e.g., reasoning about specific protein properties) and high-level thematic mastery (e.g., protein folding as a topic). This dual-level representation enables our graph to reflect both detailed scientific concepts and broader domain understanding, which is crucial for evaluating comprehensive knowledge in life sciences.
> >
> >    5. **Ablation studies to justify the design choices:** We conducted ablation experiments in the global response section, testing different parameter configurations. The results demonstrate the robustness of our evaluation method across various settings, validating our design choices.

---

> ### Author Response · Authors · 2025-11-27
> **Follow-up on responses ahead of rebuttal deadline**
>
> Dear Reviewer f3N5,
>
> We would like to express our sincere gratitude for the time and care you have devoted to reviewing our submission. Your thoughtful and constructive comments have been invaluable. We have carefully addressed each point and provided detailed responses and clarifications.
>
> As the rebuttal deadline is approaching, we would be grateful if you could kindly review the pending clarifications and share any feedback at your earliest convenience. Allowing a bit of time before the deadline would help us engage more fully with any further discussion and further improve the manuscript.
>
> We apologize for the inconvenience of this follow-up and truly appreciate your understanding, guidance, and service to the community. Thank you again for your support.
>
> With respect and appreciation,
>
> Authors

---

### Official Review · Reviewer_WF9d · 2025-10-21

**Soundness:** 2
**Presentation:** 3
**Contribution:** 4
**Rating:** 4
**Confidence:** 4

**Summary:**

The paper introduces LG-Bench, a novel, large-scale, graph-structured evaluation benchmark for LLMs in the life sciences. It comprises over 10,000 multiple-choice questions derived from recent scientific literature. The benchmark transforms this corpus into a weighted evaluation graph and introduces two new graph-based metrics—the Global Coherence Score (GCS) and Knowledge Balance Score (KBS), designed to measure structured understanding and conceptual balance beyond standard accuracy metrics.

**Strengths:**

## 1. High Novelty and Vision

- The move from flat-list evaluation to a graph-structured framework is original and timely, addressing a known limitation in LLM evaluation.

## 2. Sophisticated Data Pipeline

- The multi-stage generation and filtering pipeline—combining multiple LLMs and expert review—is impressive, though the scope of human validation remains unclear.

## 3. Potential Diagnostic Insight

- **GCS** and **KBS** are conceptually appealing and could become useful diagnostic measures if shown to align with human judgments of coherence.

**Weaknesses:**

## 1. Graph Construction Sensitivity and Excessive Density

- Average degree of ~586 suggests over-connectedness, risking that GCS/KBS reflect lexical similarity rather than meaningful conceptual coherence.
- Sensitivity to hyperparameters (γ, θ, κ) is not shown.
- **Require sensitivity analysis and justification for graph density.**

## 2. Lack of Human Validation for Metrics

- No human evaluation demonstrates that GCS/KBS correlate with expert assessments of model coherence.
- **A correlation study with expert judgment is essential.**

## 3. KBS Justification and Baseline Comparison

- The KBS amplification factor (α = 100) is arbitrary and unvalidated.
- Missing comparisons to simpler graph-aware baselines or knowledge-graph (KG) reasoning benchmarks.
- **Include ablation and KG-based comparisons (e.g., “"A scalable llm framework for therapeutic biomarker discovery: Grounding q/a generation in knowledge graphs and literature." ICLR 2025 Workshop on Machine Learning for Genomics Explorations. 2025.”).**

## 4. Circularity, Reproducibility,

- The same or similar LLMs are used for graph construction and evaluation, risking architectural bias.
- Missing implementation details (γ, θ, κ, embeddings, prompts).
- **Include full hyperparameters, model details, responsible release plan, and a formal ethics/biosecurity statement.**

**Questions:**

## 1. Validation Scope

- Did all 10,000 questions receive human validation, or was this a subset?
- What proportion were verified by experts versus filtered automatically by LLMs?

## 2. Expert Agreement

- Was inter-rater agreement measured among the four PhD reviewers?
- If not, how consistent were their judgments?

## 3. Graph Density Justification

- Why is such a high connectivity (avg. degree ≈ 586) appropriate for a domain-knowledge graph?
- What happens to GCS/KBS if the threshold θ or parameter γ is adjusted to produce sparser graphs?

## 4. Metric Interpretation

- How do **GCS** and **KBS** correlate with accuracy or with human-perceived coherence?
- Is there evidence that a higher GCS reflects better reasoning rather than answer clustering?

## 5. KBS Amplification (α)

- Why was α = 100 chosen?
- How sensitive are results to this factor?
- Would normalization or unamplified variance produce comparable results?

## 6. Knowledge Graph Baseline

- How does LG-Bench differ empirically from a traditional knowledge-graph-based evaluation (e.g., using DrugBank or Hetionet relations)?

---

> ### Author Response · Authors · 2025-11-14
> **Response to Reviewer WF9d**
>
> We thank the reviewer for the careful and constructive feedback. We are pleased that they recognized the high novelty and forward-looking vision of our work, the sophistication and rigor of our data pipeline, and the potential of GCS/KBS to diagnose structured understanding. These comments will help us further improve the paper, and we will address, point by point in the rebuttal, the key issues regarding graph density, human validation, KBS parameterization, and knowledge-graph baselines.
>
> Responses to Questions:
>
> Q1: Validation Scope
>
> A1: We first generated 13,250 questions based on life science papers. A multi-model committee then filtered out 2,653 low-quality questions (e.g., shallow, lacking textual evidence, or structurally flawed), leaving 10,597 higher-quality questions. During expert validation, we verified 4,153 questions, ultimately removing 397 questions. The final number of questions which was checked by experts is 3,756. The tables below show the details of  each stage.
>
> | Stage   | Description                                  | Number         |
> | ------- | -------------------------------------------- | -------------- |
> | Stage 1 | Initially generated question (LLM)           | 13,250         |
> | Stage 2 | After selecting by the Multi-Model Committee | 10,597 (-2653) |
> | Stage 3 | After experts review（final reservation）    | 10,200 (-397)  |
>
>
> | Error Type                         | Number |
> | ---------------------------------- | ------ |
> | Logical or factual errors          | 47     |
> | Low relevance to the domain        | 214    |
> | Topics not suitable for evaluation | 136    |
>
> Q2: Expert Agreement
>
> A2: We retained the annotation records during the review, and the calculated Fleiss' Kappa of 0.86 indicates a high level of agreement since we prepared detailed annotation guidelines to ensure consistency.
>
> Q3: Graph Density Justification
>
> A3: Unlike traditional knowledge graphs that emphasize sparse entity–relation structures, our evaluation graph models question–question connectivity to capture assessment-oriented links. We intentionally encode both low-level semantics (e.g., reasoning about specific protein properties) and high-level thematic mastery (e.g., protein folding as a topic). To better reflect the connectivity inherent in life-science knowledge, we retain low-weight edges to preserve global structure and reveal broader topical coverage; this design choice naturally results in a higher average degree.
>
> To address concerns about sensitivity, we added a series of ablations varying  hyperparamter to induce sparser graphs. Our ablation experiments are located in the global response section at the top. It shows that global coherence (GCS) stays stable, while keyword-based scores (KBS) scale predictably with graph sparsity but preserve model ranking. Importantly, all findings about domain-specific fine-tuning remain consistent. Overall, the graph structure and metrics are robust to reasonable connectivity changes and continue to effectively diagnose both local and global competencies.
>
>
> Q4: Metric Interpretation
>
> Regarding correlation with human-perceived coherence, we selected 60 question subsets for manual verification. We had models perform open-ended question answering and used human evaluators to subjectively assess reasoning capabilities to validate correlation with GCS scores and KBS. Results showed that GCS achieved a Spearman correlation coefficient of 0.95 with human ratings, while KBS reached 0.91, both demonstrating extremely high correlation. Additionally, we conducted qualitative case studies (Figure 4; Appendix A.2.2): responses from higher-GCS models exhibit fewer cross-item contradictions and more stable rationales within topical neighborhoods, aligning with expert judgments of coherence.
>
> Regarding evidence that GCS reflects reasoning rather than answer clustering, our density ablation experiments (in global response) show that different node degrees have minimal impact on GCS, indicating that GCS is actually unaffected by clustering effects and exhibits high robustness. Qualitatively, among model pairs with similar accuracy, the higher-GCS model shows more consistent, non-contradictory reasoning across related questions (Fig. 4).

---

> ### Author Response · Authors · 2025-11-14
> **Response to Reviewer WF9d (2)**
>
> Q5:  KBS Amplification ($\alpha$​​)
>
> In our formulation (8), α functions purely as a proportional scaling constant applied to the underlying variance term in KBS. When a variable is multiplied by a constant $k$, its variance scales by $k^2$. Choosing $\alpha$ = 100 therefore serves only to map the raw variance values into a more interpretable numerical range, similar to how proportions are converted into percentages for clearer presentation. This transformation is strictly linear and preserves the ordering of all models: the ranking at $\alpha$ = 1 is identical to the ranking at $\alpha$ = 100, and the relative differences between models expand uniformly. As a result, the amplification has no influence on comparative conclusions, correlation patterns, or statistical significance tests. Using normalized or unamplified variance would produce results that are mathematically equivalent up to this linear factor, and all substantive interpretations, including model ordering and domain-level insights, remain unchanged.
>
> Q6:  Knowledge Graph Baseline
>
> To empirically compare LG-Bench with traditional knowledge-graph-based pipelines, such as those using Hetionet relations, we selected relations from Hetionet and followed the approach proposed in *“A scalable LLM framework for therapeutic biomarker discovery: Grounding Q/A generation in knowledge graphs and literature”* to construct 1000 small-scale benchmark datasets (generated using the GPT-5 advanced model to ensure quality). Our experimental results show that, in terms of dataset quality, Hetionet-based datasets achieve similarly high accuracy (Acc) with only minor differences, whereas LG-Bench demonstrates a clear performance gap. This improvement is attributed to our datasets being generated from the latest high-quality literature and curated through a complex human-in-the-loop pipeline.
>
> Moreover, evaluation using graph-structured metrics produces richer and more diverse insights than Acc alone, further demonstrating the effectiveness of LG-Bench in assessing life-science knowledge and model capabilities.
>
> | Model         | Acc(Hetionet) | Acc(Ours) | GCS(Hetionet) | KBS(Hetionet) |
> | :------------ | :-----------: | :-------: | :-----------: | :-----------: |
> | Qwen2.5-14B   |     89.40     |   83.69   |     75.19     |     21.32     |
> | Qwen2.5-32B   |     89.70     |   84.87   |     75.42     |     18.19     |
> | Qwen2.5-72B   |     90.10     |   86.05   |     76.11     |     18.10     |
> | Llama-3.1-70B |     89.90     |   82.54   |     74.90     |     20.47     |
>
> After incorporating discussion and feedback, the revised version will include an expanded section comparing our pipeline with this line of KG-grounded dataset construction in a more systematic way, together with appropriate citations.
>
> Responses to Weakness:
> 1. Graph Construction Sensitivity and Excessive Density: See also in Q3.
>
> 2. Lack of Human Validation for Metrics: See also in Q2, Q4.
>
> 3. KBS Justification and Baseline Comparison: See also in Q5, Q6.
>
> 4. Circularity, Reproducibility:
>
> Details of LG-Bench have shown in Appendix A.1, including parameters, example questions, and graph structure.
>
> We promise to release the full evaluation code and all benchmark questions after completing the ethics/biosecurity review process. The specific procedure is as follows: First, we will conduct manual re-verification of ethics and biosafety-related questions, and submit our research protocol to the relevant institutional review board, including detailed documentation of data collection methods, privacy protection measures, and potential risks. Second, we will review  all benchmark questions again to ensure they do not contain harmful, biased, or sensitive content that could lead to discriminatory outcomes. Upon receiving formal ethics approval, we will make all materials publicly available to ensure the reproducibility and transparency of our research.

---

> ### Author Response · Authors · 2025-11-27
> **Follow-up on responses ahead of rebuttal deadline**
>
> Dear Reviewer WF9d,
>
> We would like to express our sincere gratitude for the time and care you have devoted to reviewing our submission. Your thoughtful and constructive comments have been invaluable. We have carefully addressed each point and provided detailed responses and clarifications.
>
> As the rebuttal deadline is approaching, we would be grateful if you could kindly review the pending clarifications and share any feedback at your earliest convenience. Allowing a bit of time before the deadline would help us engage more fully with any further discussion and further improve the manuscript.
>
> We apologize for the inconvenience of this follow-up and truly appreciate your understanding, guidance, and service to the community. Thank you again for your support.
>
> With respect and appreciation,
>
> Authors

---

### Official Review · Reviewer_zDqV · 2025-11-01

**Soundness:** 3
**Presentation:** 3
**Contribution:** 3
**Rating:** 6
**Confidence:** 3

**Summary:**

This paper introduces LG-Bench, a novel graph-structured benchmark for evaluating large language models in the life sciences. The authors argue that traditional "flat list" benchmarks fail to test an LLM's understanding of interconnected scientific concepts.

They construct the benchmark questions by creating questions from peer-reviewed scientific papers, which serve as the graph's nodes, and then connecting these nodes with weighted edges that represent the semantic similarity between the questions.

This graph structure is then used for evaluation. The paper introduces two new metrics that go beyond simple accuracy:

-Global Coherence Score (GCS): Measures a model's consistency by assessing whether it correctly answers clusters of related questions.

-Knowledge Balance Score (KBS): Analyzes the distribution of a model's errors to identify "conceptual blind spots" or uneven knowledge.

The authors show that these metrics can reveal insights into model performance, such as the trade-offs of domain-specific fine-tuning, that traditional accuracy scores cannot.

**Strengths:**

* Clear thesis and focus, as well as well-articulated findings
* As far as I am aware, their graph proposal is a novel concept for a life sciences LLM benchmark, in using graphs to identify more fine-grained nuances in LLM life sciences
* Logical argument for testing the quality of data

**Weaknesses:**

* Not thorough in assessing substitute and alternative approaches
* The authors do not describe the detailed effects (quantitatively, fine-grained breakdown) of the expert human review. There can be valuable insights here for the community (for example regarding patterns in LLM hallucination in question'

**Questions:**

Can you study alternatives more deeply?

What is the percentage of questions that were rejected or rewritten by the human experts? What are the total number of questions generated by the AI before the human validation stage?

While this appears to be a useful, unique benchmark, it is often the follow-up questions (and follow-ups to those follow-ups etc) that more akin to real life science work. Could you extend your method to evaluate this in any way?

---

> ### Author Response · Authors · 2025-11-14
> **Response to Reviewer zDqV**
>
> We sincerely thank the reviewer for their thoughtful and constructive feedback on our work. We are encouraged that the reviewer recognizes our contribution as novel and finds our graph-structured benchmark approach to be a unique concept for evaluating LLMs in life sciences. We appreciate the acknowledgment of our clear thesis, well-articulated findings, and logical argument for testing data quality. Your valuable suggestions will help us strengthen our paper significantly.
>
> Responses to Questions
>
> Q1: Can you study alternatives more deeply?
>
> A1:  To address concerns about sensitivity, we added a series of ablations varying  hyperparamter to induce sparser graphs. Our ablation experiments are located in the global response section at the top.
>
> Also to empirically compare LG-Bench with traditional knowledge-graph-based pipelines, such as those using Hetionet relations, we selected relations from Hetionet and followed the approach proposed in *“A scalable LLM framework for therapeutic biomarker discovery: Grounding Q/A generation in knowledge graphs and literature”* to construct 1000 small-scale benchmark datasets (generated using the GPT-5 advanced model to ensure quality). Our experimental results show that, in terms of dataset quality, Hetionet-based datasets achieve similarly high accuracy (Acc) with only minor differences, whereas LG-Bench demonstrates a clear performance gap. This improvement is attributed to our datasets being generated from the latest high-quality literature and curated through a complex human-in-the-loop pipeline.
>
> Moreover, evaluation using graph-structured metrics produces richer and more diverse insights than ACC alone, further demonstrating the effectiveness of LG-Bench in assessing life-science knowledge and model capabilities.
>
> | Model         | Acc(Hetionet) | Acc(Ours) | GCS(Hetionet) | KBS(Hetionet) |
> | :------------ | :-----------: | :-------: | :-----------: | :-----------: |
> | Qwen2.5-14B   |     89.40     |   83.69   |     75.19     |     21.32     |
> | Qwen2.5-32B   |     89.70     |   84.87   |     75.42     |     18.19     |
> | Qwen2.5-72B   |     90.10     |   86.05   |     76.11     |     18.10     |
> | Llama-3.1-70B |     89.90     |   82.54   |     74.90     |     20.47     |
>
> Q2: What is the percentage of questions that were rejected or rewritten by the human experts? What are the total number of questions generated by the AI before the human validation stage?
>
> A2:   We first generated 13,250 questions based on life science papers. A multi-model committee then filtered out 2,653 low-quality questions (e.g., shallow, lacking textual evidence, or structurally flawed), leaving 10,597 higher-quality questions. During expert validation, we verified  4,153 questions, ultimately removing 397 questions. The final number of questions which was checked by experts is 3,756. The tables below show the details of  each stage.
>
> | Stage   | Description                                  | Number         |
> | ------- | -------------------------------------------- | -------------- |
> | Stage 1 | Initially generated question (LLM)           | 13,250         |
> | Stage 2 | After selecting by the Multi-Model Committee | 10,597 (-2653) |
> | Stage 3 | After experts review（final reservation）    | 10,200 (-397)  |
>
>
> | Error Type                         | Number |
> | ---------------------------------- | ------ |
> | Logical or factual errors          | 47     |
> | Low relevance to the domain        | 214    |
> | Topics not suitable for evaluation | 136    |

---

> ### Author Response · Authors · 2025-11-14
> **Response to Reviewer zDqV (2)**
>
> Q3: While this appears to be a useful, unique benchmark, it is often the follow-up questions (and follow-ups to those follow-ups etc) that more akin to real life science work. Could you extend your method to evaluate this in any way?
>
> A3: This is an excellent point that aligns with real scientific inquiry. We propose the following extensions:
>
> For follow-up questions regarding continuous issues, the approach demonstrated on the graph structure involves starting from one node and performing a breadth-first search outward until the product of the weights along the path falls below a certain threshold (0.15). We randomly sampled 100 of these paths, and for each path, we calculated the accuracy as the longest length of correctly predicted prefix paths divided by the path length. Then, we computed the average of all path accuracies.
>
> For a path formalized as $u_0, u_1, \ldots, u_k $, the correctness condition can be defined using a result function $\text{Res}_M(u) $. For a prefix path$u_0, u_1, \ldots, u_t (t \leq k)$ to be considered correct, the following condition must be met:
>
> $$
> \text{Res}_M(u_0) = 1, \text{Res}_M(u_1) = 1, \ldots, \text{Res}_M(u_t) = 1
> $$
>
> We consider that the length of the longest correctly predicted prefix path indicates the quality of the answer to the follow-up chain of questions. So accuracy calculation for each path is computed  as:
>
> $$
> \text{Accuracy}_{\text{path}} = \frac{\text{Length of longest correctly predicted prefix path}}{\text{Length of total path}}
> $$
>
> Average Accuracy: Compute the mean accuracy across all sampled paths:
>
> $$
> \text{Acc}\_{follow} = \frac{1}{N} \sum_{i=1}^{N} \text{Accuracy}_{\text{path}_i}
> $$
>
> where $ N $ is the number of sampled paths.
>
> Here is the result:
>
> | Model                 | Qwen2.5-0.5B | Qwen2.5-7B | Qwen2.5-14B | Qwen2.5-32B | Qwen2.5-72B |
> | --------------------- | ------------ | ---------- | ----------- | ----------- | ----------- |
> | $\text{Acc}\_{follow}$ | 51.87%       | 64.32%     | 69.23%      | 69.83%      | 76.37%      |
>
> We observe that for LG-Bench, the graph structure can also effectively reflect their reasoning capabilities for follow-up questions in the life science domain, which demonstrates the potential of our graph structure.
>
> Responses to Weakness:
> 1. Not thorough in assessing substitute and alternative approaches: See also in Q1, Q3.
>
> 2. The authors do not describe the detailed effects (quantitatively, fine-grained breakdown) of the expert human review. There can be valuable insights here for the community (for example regarding patterns in LLM hallucination in question: See also in Q2.

---

> ### Author Response · Authors · 2025-11-27
> **Follow-up on responses ahead of rebuttal deadline**
>
> Dear Reviewer zDqV,
>
> We would like to express our sincere gratitude for the time and care you have devoted to reviewing our submission. Your thoughtful and constructive comments have been invaluable. We have carefully addressed each point and provided detailed responses and clarifications.
>
> As the rebuttal deadline is approaching, we would be grateful if you could kindly review the pending clarifications and share any feedback at your earliest convenience. Allowing a bit of time before the deadline would help us engage more fully with any further discussion and further improve the manuscript.
>
> We apologize for the inconvenience of this follow-up and truly appreciate your understanding, guidance, and service to the community. Thank you again for your support.
>
> With respect and appreciation,
>
> Authors

---

### Author Response · Authors · 2025-11-14
**Global Response (1)**

All three reviewers had questions about the density comparison and parameter selection of the graph structure. To address these concerns, we conducted ablation experiments with different parameter ($\theta$ and $\gamma$) to demonstrate the robustness of our approach.

We observe that GCS remains essentially stable across settings, indicating that global coherence is not an artifact of density. KBS magnitudes change almost multiplicatively with sparsity, which is expected given its variance-like nature, reducing edge counts inflates the variance term, but the rank ordering of models is consistent across nearly all subdomains and in the overall benchmark. Crucially, all conclusions about domain-specific fine-tuning remain unchanged. These results suggest that our graph is appropriate for diagnosing both local and global competencies, while our metrics are robust to reasonable changes in connectivity.

Each table is set with different $\theta$. For each $\theta$, the parameters GCS, KBS, and node degree are reported using $\gamma$ = 0.4 ($\gamma$ = 0.6) to indicate the comparison between $\gamma$ values. For example, GCS 14.04 (14.37) means that GCS is 14.04 when $\gamma$ = 0.4 and 14.37 when $\gamma$ = 0.6.

1. When $\theta = 0.45$, Avg. Degree = 249.61 (257.42).

|      MODEL      |       |   MEDICINE    |               |       |    BIOLOGY    |               |       |   CHEMISTRY   |               |       |    OVERALL    |               |
| :-------------: | :---: | :-----------: | :-----------: | :---: | :-----------: | :-----------: | :---: | :-----------: | :-----------: | :---: | :-----------: | :-----------: |
|                 |  Acc  |      GCS      |      KBS      |  Acc  |      GCS      |      KBS      |  Acc  |      GCS      |      KBS      |  Acc  |      GCS      |      KBS      |
|   Qwen2.5-7B    | 78.38 | 62.59 (62.05) | 60.38 (61.25) | 80.03 | 64.29 (64.72) | 40.12 (40.89) | 76.76 | 59.29 (59.81) | 48.07 (48.94) | 78.43 | 62.20 (62.68) | 29.83 (30.56) |
|   Llama-3-8B    | 72.77 | 54.41 (54.86) | 69.05 (70.01) | 75.67 | 57.68 (58.15) | 32.90 (33.72) | 72.53 | 53.21 (53.74) | 52.91 (53.85) | 73.73 | 55.25 (55.78) | 29.14 (29.96) |
| Llama3-Med42-8B | 78.10 | 61.79 (62.31) | 55.59 (55.52) | 77.38 | 60.17 (60.64) | 35.06 (35.91) | 75.00 | 56.82 (57.38) | 55.66 (56.73) | 76.82 | 59.59 (60.12) | 27.72 (28.45) |
|   Qwen2.5-14B   | 83.02 | 69.84 (70.29) | 43.47 (44.38) | 85.26 | 72.75 (73.21) | 36.85 (37.69) | 82.62 | 68.81 (69.35) | 38.89 (39.82) | 83.69 | 70.56 (71.09) | 22.18 (22.91) |
|   Qwen2.5-32B   | 84.70 | 72.79 (73.26) | 44.72 (45.67) | 86.06 | 74.41 (74.94) | 29.19 (30.03) | 83.75 | 70.58 (71.13) | 38.19 (39.14) | 84.87 | 72.78 (73.32) | 21.94 (22.68) |
|  Llama-3.1-70B  | 82.77 | 69.74 (70.18) | 48.59 (49.51) | 83.14 | 69.25 (69.76) | 38.64 (39.58) | 81.67 | 66.95 (67.49) | 45.48 (46.42) | 82.54 | 68.75 (69.28) | 25.85 (26.67) |
|   Qwen2.5-72B   | 86.26 | 75.37 (75.83) | 38.76 (39.68) | 86.97 | 75.76 (76.28) | 29.24 (30.16) | 84.85 | 72.67 (73.21) | 41.59 (42.54) | 86.05 | 74.73 (75.27) | 20.73 (21.55) |

2. When  $\theta = 0.5$, Avg. Degree = 110.75 (119.17).

|      MODEL      |       |   MEDICINE    |                 |       |    BIOLOGY    |                |       |   CHEMISTRY   |                 |       |    OVERALL    |               |
| :-------------: | :---: | :-----------: | :-------------: | :---: | :-----------: | :------------: | :---: | :-----------: | :-------------: | :---: | :-----------: | :-----------: |
|                 |  Acc  |      GCS      |       KBS       |  Acc  |      GCS      |      KBS       |  Acc  |      GCS      |       KBS       |  Acc  |      GCS      |      KBS      |
|   Qwen2.5-7B    | 78.38 | 62.63 (63.12) | 124.92 (127.43) | 80.03 | 64.11 (64.58) | 77.40 (79.62)  | 76.78 | 59.29 (59.78) |  89.46 (91.77)  | 78.43 | 62.23 (62.71) | 64.70 (66.21) |
|   Llama-3-8B    | 72.77 | 54.40 (54.89) | 142.21 (145.09) | 75.67 | 57.50 (57.96) | 61.07 (63.29)  | 72.55 | 53.16 (53.67) | 99.42 (101.82)  | 73.73 | 55.32 (55.83) | 63.26 (65.35) |
| Llama3-Med42-8B | 78.10 | 61.78 (62.28) | 117.57 (119.93) | 77.38 | 60.14 (60.62) | 63.90 (66.27)  | 75.02 | 56.87 (57.38) | 104.30 (106.81) | 76.82 | 59.64 (60.15) | 58.67 (60.47) |
|   Qwen2.5-14B   | 83.02 | 69.64 (70.14) |  95.62 (97.81)  | 85.26 | 72.70 (73.18) | 60.96 (63.11)  | 82.64 | 68.92 (69.44) |  71.85 (74.07)  | 83.69 | 70.60 (71.11) | 44.11 (46.00) |
|   Qwen2.5-32B   | 84.70 | 72.59 (73.09) | 101.22 (103.50) | 86.06 | 74.37 (74.87) | 57.84 (59.92)  | 83.77 | 70.63 (71.15) |  71.71 (71.70)  | 84.87 | 72.81 (73.33) | 45.99 (47.81) |
|  Llama-3.1-70B  | 82.77 | 69.62 (70.11) | 106.73 (109.12) | 83.14 | 69.20 (69.71) | 77.25 (79.43)  | 81.69 | 67.01 (67.54) |  83.49 (85.32)  | 82.54 | 68.81 (69.34) | 53.04 (55.41) |
|   Qwen2.5-72B   | 86.26 | 75.14 (75.64) |  89.02 (91.17)  | 86.97 | 75.71 (76.21) | 57.89 (60.03)  | 84.88 | 72.77 (73.30) |  77.55 (79.23)  | 86.05 | 74.78 (75.31) | 43.04 (45.21) |

---

> ### Author Response · Authors · 2025-11-14
> **Global Response (2)**
>
> 3. When $\theta = 0.6$, Avg. Degree = 23.98 (24.12).
>
> |      MODEL      |       |   MEDICINE    |                 |       |    BIOLOGY    |                 |       |   CHEMISTRY   |                 |       |    OVERALL    |                 |
> | :-------------: | :---: | :-----------: | :-------------: | :---: | :-----------: | :-------------: | :---: | :-----------: | :-------------: | :---: | :-----------: | :-------------: |
> |                 |  Acc  |      GCS      |       KBS       |  Acc  |      GCS      |       KBS       |  Acc  |      GCS      |       KBS       |  Acc  |      GCS      |       KBS       |
> |   Qwen2.5-7B    | 78.44 | 61.94 (62.48) | 262.53 (261.28) | 80.00 | 63.79 (64.21) | 155.60 (156.73) | 76.74 | 58.74 (58.95) | 215.85 (217.12) | 78.43 | 62.09 (62.84) | 134.09 (135.38) |
> |   Llama-3-8B    | 72.73 | 53.61 (53.28) | 301.92 (300.45) | 75.67 | 57.15 (56.89) | 125.30 (126.58) | 72.50 | 52.81 (53.47) | 242.89 (241.73) | 73.73 | 55.14 (55.92) | 139.71 (140.86) |
> | Llama3-Med42-8B | 78.10 | 60.97 (61.73) | 259.10 (258.15) | 77.35 | 59.88 (59.56) | 127.72 (128.91) | 74.98 | 56.51 (57.12) | 272.32 (271.48) | 76.82 | 59.51 (60.35) | 129.37 (130.62) |
> |   Qwen2.5-14B   | 83.07 | 68.98 (69.65) | 218.37 (217.24) | 85.24 | 72.40 (72.78) | 129.84 (131.15) | 82.61 | 68.34 (68.92) | 179.09 (180.38) | 83.69 | 70.47 (71.23) |  95.29 (96.78)  |
> |   Qwen2.5-32B   | 84.69 | 72.01 (71.58) | 219.29 (220.46) | 86.07 | 74.18 (74.83) | 115.46 (116.72) | 83.75 | 70.21 (70.94) | 168.08 (169.54) | 84.87 | 72.76 (73.42) |  95.15 (96.19)  |
> |  Llama-3.1-70B  | 82.76 | 68.95 (68.47) | 228.51 (227.38) | 83.12 | 68.95 (69.71) | 155.13 (156.47) | 81.66 | 66.78 (67.35) | 198.68 (199.82) | 82.54 | 68.82 (69.58) | 111.08 (112.35) |
> |   Qwen2.5-72B   | 86.25 | 74.32 (74.86) | 201.61 (200.73) | 86.95 | 75.40 (75.92) | 119.89 (121.24) | 84.85 | 72.36 (72.98) | 180.75 (181.93) | 86.05 | 74.65 (75.38) |  91.82 (93.15)  |

---

### Author Response · Authors · 2025-11-26
**Follow-up on responses ahead of rebuttal deadline**

Dear Area Chairs and Reviewers,

We would like to express our sincere gratitude for the time and care you have devoted to reviewing our submission. Your thoughtful and constructive comments have been invaluable. We have carefully addressed each point and provided detailed responses and clarifications.

As the rebuttal deadline is approaching, we would be grateful if you could kindly review the pending clarifications and share any feedback at your earliest convenience. Allowing a bit of time before the deadline would help us engage more fully with any further discussion and further improve the manuscript.

We apologize for the inconvenience of this follow-up and truly appreciate your understanding, guidance, and service to the community. Thank you again for your support.

With respect and appreciation,

Authors

---

### Author Response · Authors · 2025-12-02
**Summary for Area Chair**

We sincerely thank the Area Chair and the reviewers for their time and constructive feedback. We are encouraged that the reviewers recognize the novelty and significance of our work. During the rebuttal phase, we conducted extensive additional experiments (including graph density ablations, human-metric correlation studies, and baseline comparisons) and provided detailed analyses. We believe these efforts have effectively addressed the reviewers' concerns and further strengthened the paper.

**A. Paper Overview**

LG-Bench introduces a novel graph-structured benchmark with over 10,000 multiple-choice questions for evaluating LLMs in life sciences. Unlike traditional "flat list" benchmarks, our approach organizes questions into a weighted evaluation graph that captures conceptual relationships. This structure enables two new diagnostic metrics: Global Coherence Score (GCS) for measuring consistency across related questions, and Knowledge Balance Score (KBS) for identifying conceptual blind spots.

**B. Reviewer Consensus**

All three reviewers highlighted the high novelty and forward-looking vision of our graph-based framework:

*   Reviewer zDqV: " Clear thesis and focus, as well as well-articulated findings. Novel concept for a life sciences LLM benchmark"
*   Reviewer WF9d: "High Novelty and Vision... original and timely... sophisticated data pipeline."
*   Reviewer f3N5: "Novel and can inspire future research to gauge models on different levels of topology."

The reviewers also consistently praised our sophisticated data pipeline, clear presentation, and comprehensive experiments.

**C. Key Concerns Addressed**

a. Graph Density and Hyperparameter Sensitivity (Reviewer WF9d, zDqV, f3N5)

We conducted extensive ablation experiments varying $\gamma$ and $\theta$ (see Global Response) to produce graphs with significantly different densities (avg. degree ranging from ~250 down to ~24). Results demonstrate the robustness of our approach:

*   GCS Stability: GCS scores remain stable across all settings, proving it is not an artifact of density.
*   KBS Consistency: While KBS magnitudes scale predictably with sparsity (due to its variance-based nature), the rank ordering of models remains consistent.
*   All insights regarding domain-specific fine-tuning hold true regardless of graph density.

b. Human Validation and Expert Agreement (Reviewer WF9d, zDqV, f3N5)

*   Pipeline Rigor: We clarified the funnel: 13,250 initial questions $\rightarrow$ 10,597 after multi-model filtering $\rightarrow$ 10,200 final (3,756 expert-verified).
*   Expert Agreement: We reported a Fleiss' Kappa of 0.86 among four expert reviewers, indicating high inter-rater reliability.
*   Metric Correlation: To validate GCS and KBS, we conducted a new human evaluation on a subset of 60 questions. Human evaluators assessed the coherence of model reasoning. Results show extremely high correlation: GCS achieved a Spearman correlation of 0.95 with human judgments, and KBS achieved 0.91.

c. Comparison with Knowledge Graph Baselines (Reviewer zDqV, WF9d)

We compared against Hetionet-based benchmarks following prior work (A scalable llm framework for therapeutic biomarker discovery: Grounding q/a generation in knowledge graphs and literature. ICLR 2025 Workshop on Machine Learning for Genomics Explorations. 2025). LG-Bench shows clearer performance differentiation (83.69% - 86.05% (LG-Bench) vs 89.40% - 90.10% (Hetionet-based benchmark) for Qwen2.5-14B - Qwen2.5-72B), attributed to our literature-grounded, human-curated pipeline. Also, Graph-structured metrics provide richer diagnostic insights than accuracy alone.

d. Follow-up Question Evaluation (Reviewer zDqV)

We extended our method to evaluate chain-of-questions using BFS paths on the graph, demonstrating that our structure effectively captures reasoning capabilities for sequential scientific inquiry. Details are described in the A3 part of Response to Reviewer zDqV (2).

e. Other Technical Clarifications (Reviewer f3N5, WF9d)
We addressed specific technical questions to ensure reproducibility and clarity:

*   Definitions: Clarified the definition of "weighted accuracy" and the rationale for the KBS amplification factor ($\alpha$) as a linear scaling constant for interpretability.
*   Design Choices: Explained the keyword extraction strategy and the multi-granular representation choice.
*   Reproducibility: Provided details on the prompts used for generation/assessment and the specific models (Claude-Opus-4, GPT-4o) employed in the pipeline. We have committed to releasing the full code, prompts, and benchmark data.

**D. Significance**

Our metrics reveal insights invisible to traditional accuracy, such as how domain-specific fine-tuning can improve local performance while potentially disrupting global coherence. We believe this contribution opens a new direction for model evaluation.

---

### Meta-Review · Area_Chair_DgNK · 2026-01-07

**Summary:**

Based on a comprehensive review of the paper, the reviewer comments, and the authors' rebuttal, my recommendation is to **Reject** this submission. However, I strongly **encourage the authors to recycle** this work for a future submission, as it holds promise but requires significant maturation to meet the bar for a top-tier general machine learning conference.

Below is the summary of the comments and the specific rationale for this decision.

### Summary of Comments

The submission proposes **LG-Bench**, a graph-structured benchmark for life sciences containing roughly 10,000 questions. The core novelty is using a "weighted evaluation graph" to calculate Global Coherence Scores (GCS) and Knowledge Balance Scores (KBS) rather than simple accuracy.

*   **Reviewer zDqV (Score: 6):** Found the graph concept novel and the thesis clear. However, they noted a lack of thorough assessment of alternative approaches and requested more quantitative details regarding the expert review process and rejection rates.
*   **Reviewer WF9d (Score: 4):** Provided the most critical assessment. While acknowledging the data pipeline's sophistication, they raised significant concerns about **graph density** (average degree ~586), arguing this might reflect lexical similarity rather than conceptual coherence. They strongly criticized the lack of human validation for the proposed metrics and found the KBS amplification factor ($\alpha=100$) arbitrary.
*   **Reviewer f3N5 (Score: 6):** Praised the writing and the "convolution" evaluation method. However, they pointed out that key definitions (like weighted accuracy) were unclear and that the motivation for specific design choices (keyword extraction granularity) was missing.

**Authors' Response:**
In their rebuttal, the authors conducted ablation studies to show metrics remained stable at lower graph densities, added a human validation study on a small subset (60 questions) showing high correlation, and compared their work against a Hetionet-based baseline.

### Rationale for Rejection

Despite the authors' extensive rebuttal efforts, several foundational issues suggest this paper is not yet ready for publication at this venue.

**1. Benchmark Scale and Scope**
For a primary benchmark paper in 2026, a dataset of **10,000 questions** is relatively small, especially for a domain as vast as Life Sciences (Medicine, Biology, Chemistry). While the "expert curation" is a strength, the scale limits the benchmark's ability to serve as a definitive standard for large foundation models. The paper frames itself as a broad benchmark, but the volume of data does not match the breadth of the claims.

**2. Validity of the Graph Construct**
Reviewer WF9d’s concern regarding the **high graph density** (Average Degree ~586) remains a critical flaw in the design philosophy. While the authors provided ablation studies in the rebuttal, the fundamental question remains: distinct "reasoning paths" are difficult to isolate in such a densely connected graph. There is a lingering risk that the GCS metric measures a model's ability to recall clusters of keywords rather than genuine logical consistency. The graph construction relies heavily on semantic similarity of keywords, which can conflate "related topics" with "logical dependencies."

**3. "Afterthought" Validation**
Crucial validation steps—specifically the comparison to existing Knowledge Graph (KG) baselines (like Hetionet) and human correlation studies—were only introduced in the **rebuttal phase**.
*   **Human Eval:** The human validation was performed on only **60 questions**. For a benchmark of 10,000 items claiming to introduce a novel metric (GCS), validating on 0.6% of the dataset is statistically weak.
*   **Baselines:** The comparison to standard KG methods should be central to the main paper to justify the novel graph format, not an appendix added to satisfy reviewers.

**4. Arbitrary Design Choices**
The selection of hyperparameters, such as the KBS amplification factor $\alpha=100$, appears arbitrary. The authors explained this is for "interpretability", but this suggests the raw metric naturally produces very small, hard-to-read variances, raising questions about its sensitivity without artificial scaling.

### Encouragement to Recycle

The authors are **strongly encouraged to recycle** this work. The core idea—moving from "flat" accuracy to "topological" consistency—is technically interesting and diagnostically valuable.

**Recommendations for the next version:**
1.  **Integrate Rebuttal Data:** Move the KG baseline comparisons and density ablation studies from the rebuttal into the main body of the paper. These are essential for justifying the method.
2.  **Expand Validation:** Conduct a much larger human evaluation study (beyond 60 questions) to prove that GCS actually aligns with human perceptions of "coherence."
3.  **Clarify the "Why":** Better articulate *why* the community needs a graph-structured evaluation over simply grouping questions by fine-grained topics. The distinction between "answering a cluster of questions" vs. "reasoning over a graph" needs to be sharper.
4.  **Target Evaluation-Specific Venues:** This paper might find a better home in venues specifically focused on NLP evaluation methodologies or Biomedical AI, where the diagnostic utility of the graph structure can be appreciated even with a smaller dataset size.

**Reviewer Concerns:**

*   **Reviewer zDqV (Score: 6):** Found the graph concept novel and the thesis clear. However, they noted a lack of thorough assessment of alternative approaches and requested more quantitative details regarding the expert review process and rejection rates.

*   **Reviewer WF9d (Score: 4):** Provided the most critical assessment. While acknowledging the data pipeline's sophistication, they raised significant concerns about **graph density** (average degree ~586), arguing this might reflect lexical similarity rather than conceptual coherence. They strongly criticized the lack of human validation for the proposed metrics and found the KBS amplification factor ($\alpha=100$) arbitrary.

*   **Reviewer f3N5 (Score: 6):** Praised the writing and the "convolution" evaluation method. However, they pointed out that key definitions (like weighted accuracy) were unclear and that the motivation for specific design choices (keyword extraction granularity) was missing.

**Reviewer Scores:**

Nothing changed.

---

### Decision · Program_Chairs · 2026-01-26

Reject